# Understanding the Toxin Effects of β-Zearalenol and HT-2 on Bovine Granulosa Cells Using iTRAQ-Based Proteomics

**DOI:** 10.3390/ani10010130

**Published:** 2020-01-13

**Authors:** Lian Li, Min Yang, Chengmin Li, Fangxiao Yang, Genlin Wang

**Affiliations:** College of Animal Science and Technology, Nanjing Agricultural University, Nanjing 210095, China; 2018105029@njau.edu.cn (M.Y.); 2016205003@njau.edu.cn (C.L.); 2016105024@njau.edu.cn (F.Y.); glwang@njau.edu.cn (G.W.)

**Keywords:** bovine, β-zearalenol, HT-2, differentially expressed proteins, granulosa cells

## Abstract

**Simple Summary:**

Zearalenone (ZEA) and T-2 are two important mycotoxins, which have deleterious effects on the health of humans and livestock. ZEA and its derivatives, α-zearalenol and β-zearalenol, disturb the hormonal homeostasis and lead to numerous problems in the reproductive system. The HT-2 toxin, as the primary metabolite of the T-2 toxin, exerts a series of toxic effects on humans and livestock. The T-2 toxin and its metabolite HT-2 toxin induce damages in multiple tissues, which include the reproductive system. However, toxic response profiles of these mycotoxins on bovine ovarian granulosa cells (bGCs) are unclear. In this study, we determined the importance of heat shock proteins, clarified oxidative stress, and the caspase-3 signaling cascade involved in the mycotoxin-treated toxic response. These results could provide new insights for future studies on prevention and treatment of reproductive problems caused by mycotoxins in bovines.

**Abstract:**

Zearalenone (ZEA) and T-2 are the most common mycotoxins in grains and can enter the animal and human food-chain and cause many health disorders. To elucidate the toxic response profile, we stimulated bovine granulosa cells (GCs) with β-zearalenol or HT-2. Using isobaric tags for relative and absolute quantification (iTRAQ)-based proteomic, 178 and 291 differentially expressed proteins (DEPs, fold change ≥ 1.3 and *p*-value < 0.05) in β-zearalenol and HT-2 groups were identified, respectively. Among these DEPs, there were 66 common DEPs between β-zearalenol and HT-2 groups. These 66 DEPs were associated with 23 biological processes terms, 14 molecular functions terms, and 19 cellular components terms. Most heat shock proteins (HSPs) were involved in the toxic response. Reactive oxygen species accumulation, the endoplasmic reticulum (ER)-stress related marker molecule (GRP78), and apoptosis were activated. β-zearalenol and HT-2 inhibited oestradiol (E_2_) production. These results emphasized the important function of HSPs, clarified oxidative stress, and demonstrated the caspase-3 signaling cascade involved in mycotoxin-treated toxic response, along with decreased E_2_ production. This study offers new insights into the toxicity of β-zearalenol and HT-2 on ovarian granulosa cells.

## 1. Introduction

Mycotoxins are natural compounds produced as secondary metabolites by filamentous fungi or molds, such as *Aspergillus, Fusarium, Penicillium*, and *Alternaria* spp. The mycotoxins commonly found on corn or small grains in the temperate climate zone, and their prevalence may increase because of global warming [1,2]. Zearalenone (ZEA) and T-2 are the most common mycotoxins in cereals, and they may cause several health disorders by entering the food chain of livestock and humans. ZEA, α-zearalenol, and β-zearalenol have a structural analogy to estrogen; thus, they can bind to endoplasmic reticulum (ER). These bindings disturb the hormonal homeostasis and lead to numerous problems of the reproductive system [3,4]. The T-2 toxin, a common trichothecene mycotoxin, is produced by various *Fusarium* species in contaminated crops, whereas its primary metabolite product is HT-2 toxin [5]. Wu et al. reported that the T-2 toxin can be rapidly metabolized to HT-2 toxin in microsomes of some animals’ tissues and the conversion rates are 80% [6]. The HT-2 toxin has exerted various toxic effects on multiple systems, such as digestive [7], immune [8], and reproductive systems [9]. Previous studies reported that ZEA and T-2 impaired mammalian oocyte quality [9,10,11]. However, toxic response mechanisms of these mycotoxins on ovary function are unclear.

ZEA and T2 are capable of causing reproductive disorders [10,11,12,13]. This study was performed to clarify the ZEA and T2 toxic response mechanism on ovarian function. Ovarian follicle development is a complex process. The ovarian follicular growth mainly depends on the development of the follicular granulosa cells (GCs) and oocytes in early stages, which communicate and support one another [14]. GCs play a critical role in steroid secretion, and these steroids are essential to the function and normal development of many organs [15]. The isolated GCs (from bovine ovarian follicles) and cultured in vitro is a reliable in vitro model, it has been widely used in reproductive toxicology research. The in vitro system has been successfully utilized as an in vitro model for the determination of the contaminants causing effects on steroidogenesis [16]. Evidence indicates that disordered ovulatory and ovarian function may be associated with GC dysfunctions. [17].

In this study, we profiled the effects of β-zearalenol or HT-2 on bovine GCs using an isobaric tags for relative and absolute quantification (iTRAQ) technique to obtain GCs proteomes and identify functional proteins involved in the toxin response. In particular, we evaluated the in vitro effects of β-zearalenol or HT-2 on reactive oxygen species (ROS) accumulation, steroid production, and cell apoptosis using primary bovine GCs. The interesting data resulting from these analyses will shed light on the toxic mechanisms of mycotoxins; thus, potentially further providing a theoretical basis for the prevention and treatment of reproductive problems caused by mycotoxins.

## 2. Materials and Methods

The experimental protocol was approved by the Animal Care and Use Committee of the College of Animal Science and Technology (Approval number: SYXK (SU) 2017-0027), Nanjing Agriculture University, Nanjing, China.

### 2.1. Isolation, Culture, and Identification of Bovine GCs

Healthy ovaries from non-pregnant Holstein dairy cows were acquired from a nearby slaughterhouse. The ovaries (from nine individual animals, number of ovaries = 18) were chosen and kept on ice in saline until the GCs were collected. GCs from small (1–5 mm) follicles were isolated as previously described [18]. This size classification was based on previous studies showing that small follicles (1–5 mm) are less responsive to follicle-stimulating hormone (FSH) and insulin-like growth factor 1 (IGF-1) than large (8–22 mm) follicles are [19], and that large follicles have much greater estradiol (E_2_) concentrations than small follicles [20]. Firstly, the follicular fluid was obtained from small follicles (1–5 mm) by a sterile syringe, and then centrifuged (800× *g* 5 min), then, the cells were washed twice with PBS and resuspended with medium. Finally, cell viability was determined by the trypan blue dye exclusion method and the viable cells (1.0 × 10^5^) were placed in 24-well Falcon multiwell plates in 1 mL of DMEM/Ham’s F-12 (1:1) basal medium (Gibco, Grand Island, NY, USA), containing 10% fetal bovine serum (Gibco, Gaithersburg, MD, USA) and antibiotics (Gibco, Grand Island, NY, USA). These GCs were kept at 38.5 °C in 5% CO_2_ environment and the medium was replaced by fresh medium every 24 h [10]. GCs were identified by follicle-stimulating hormone receptor (FSHR) antibody (PA5-28764, Thermo Fisher Scientific, Waltham, MA, USA) using immunocytochemistry, as described in a previous study [21].

### 2.2. Cell Treatment and Cell Viability Analysis

Cell treatment: GCs were washed twice with serum-free medium and β-zearalenol, or HT-2 treatment containing 500 ng/mL testosterone (as an oestradiol precursor) for 24 h. β-zearalenol (Sigma-Aldrich, #Z2000, St. Louis, MO, USA) and HT-2 (Sigma-Aldrich, #34136, St. Louis, MO, USA) were dissolved with DMSO and diluted with DMEM/Ham’s F-12, respectively. The treatment concentration ranges of β-zearalenol and HT-2 toxin were 0–200 μM and 0–200 nM, respectively. Doses of β-zearalenol and HT-2 were selected based on previous studies [10,13,21].

Cell viability analysis: the proliferation of bovine GCs was detected using CCK8 (Beyotime Biotechnology, Shanghai, China). GCs were treated with either β-zearalenol or HT-2 toxin for 24 h. Then, 10 μL of the CCK8 reagent was added into each well and the GCs were incubated at 25 °C for 4 h. The absorbance values were measured at 450 nm. 

According to the cell viability results, the dose of β-zearalenol and HT-2 in the following experiment was 25 μM and 50 nM, respectively. After 24 h of β-zearalenol or HT-2 treatment, the GCs were harvested for protein extraction and the culture medium was collected for subsequent determination of the concentration of hormone.

### 2.3. Protein Extraction, Digestion, and iTRAQ Labeling

GCs were lysed in the presence of 0.1% Triton X-100 and the lysates were centrifuged at 13,000 rpm for 15 min at 4 °C, and the total protein concentration was determined using a BCA protein assay kit (Beyotime Biotechnology, Shanghai, China). To digest the protein, 50 μg of protein from each sample was denatured and incubated with 10 mM dithiothreitol (DTT; Amresco, OH, USA) for 30 min at 56 °C. For protein alkylation, protein was incubated with 55 mM iodoacetamide for 40 min at 25 °C in the dark, followed by the digestion with 1 μg trypsin for 12 h at 37 °C. After the tryptic digestion, peptides were labeled according to the instructions of iTRAQ Reagent-8 plex Multiples kit (AB SCIEX, London, UK). Next, the labeled samples were fractionated using a high-performance liquid chromatography (HPLC) system (Thermo Dinoex Ultimate 3000 BioRS, Rockford, IL, USA) equipped with a Durashell C18 (5 mm, 100 Å, 4.6 × 250 mm) column. A total of 12 fractions were collected and vacuum dried.

### 2.4. Liquid Chromatography-Tandem Mass Spectrometry (LC-MS/MS) Analysis

The sample fractions described above were further separated and identified on an AB SCIEX nanoLC-MS/MS system (Triple TOF 5600 plus, Framingham, MA, USA). Mobile phase A (5% acetonitrile + 0.1% formic acid) and Mobile phase B (95% acetonitrile + 0.1% formic acid) were the mobile phase. After resuspension with buffer A, 10 μL of sample supernatant was loaded by the auto-sampler onto a C18 trap column and then separated on the reversed-phase Magic C18. The samples were loaded at 8 μL/min for 4 min, then a 44 min gradient was run at 300 nL/min starting from 2% to 35% buffer B, followed by 2 min linear gradient to 80% buffer B, and then maintenance at 80% buffer B for 4 min, and finally returned to 5% buffer B in 1 min.

The peptides were then separated using a nanocolumn (inner diameter of 3 µm). The eluted peptides were detected by LC apparatus connected to the mass spectrometer (AB SCIEX, Framingham, MA, USA) in a positive ion mode. The separated peptide fragments were identified using a mass spectrometer in positive ion mode with electrospray ionization and collision-induced dissociation (CID). The collected ranges of full-scan MS1 spectra and full-scan MS2 spectra were 360–1460 *m*/*z* for 250 ms and 50–2000 *m*/*z* for 100 ms, respectively. For MS1, the 20 most intense precursors with charge state 2–5 were selected for fragmentation. For MS2, the precursor ions were excluded from reselection for 15 s.

### 2.5. Protein Identification and Quantitative Analysis

The original file data were analyzed by the ProteinPilot Software (AB SCIEX, Framingham, MA, USA). These data were integrated into ProteinPilot for identification protein, and ProteinPilot was performed against the uniprot database for database searching. For protein identification, the parameters were set the same as in the study by Dai et al. [22]. The peptide length was set to >4, and peptide filter with FDR of 1% was used [22]. Only proteins with unused value more than 1.3 were considered for identification. 

### 2.6. Biological Function Enrichment Analysis

We conducted the functional enrichment of Gene Ontology (GO) terms and Kyoto Encyclopedia of Gene and Genomes analysis (KEGG) pathways for differentially expressed proteins (DEPs) in the DAVID database (https://david.ncifcrf.gov/). *p*-values < 0.05 were considered to be significantly enriched in GO terms and KEGG pathway analysis.

### 2.7. Determination of Hormone

The concentration of 17β-estradiol in culture medium was analyzed via radioimmunoassay (RIA) as previously described [23]. Intra- and interassay coefficients for 17β-estradiol were 8% and 13%. Steroid production was expressed as ng or pg/10^5^ cells per 24 h and cell numbers at the termination of the experiment were used for this calculation [16].

### 2.8. Western Blotting

The equal amounts of protein (50 µg) were subjected to gel electrophoresis (8% SDS-PAGE) followed by electrotransfer to a nitrocellulose membrane. The membranes were blocked by placing in blocking solution for 2 h at RT. Then the membranes were incubated overnight with diluted primary antibodies (rabbit anti-rat HSP27 (1:1000, Novus, Littleton, CO, USA), HSP70 (1:1000, Novus, Littleton, CO, USA), HSP90 (1:1000, Abcam, Cambridge, UK), HMGCS1 (1:1000, Abcam, Cambridge, UK) or FASN (1:1000, Cell Signaling Technology, Boston, MA, USA)). After washing three times with Tris-Buffered Saline Tween-20 (TBST), the membranes were incubated with peroxidase-conjugated secondary antibody (1:2000; Proteintech, Chicago, IL, USA) at 37 °C for 1 h. After wash by TBST, the membrane was visualized using ECL Plus reagent (Beyotime Biotechnology, Shanghai, China). In all analyses, β-actin was as a loading control. Densitometric analysis of proteins were performed using ImageJ software (National Institutes of Health, Bethesda, MD, USA). 

### 2.9. Apoptosis Detection by Flow Cytometry and Caspase-3 Activity Measurement

To determine the effect of β-zearalenol or HT-2 on apoptosis, the apoptosis level of GCs was detected using PE Annexin V Apoptosis Detection Kit I (BD Biosciences, San Jose, CA, USA). The rate of apoptosis was calculated by flow cytometry scatterplot.

To obtain the activity of caspase-3, the supernatants of the protein lysates in GCs were quantified for caspase-3 enzymatic activity using a Caspase 3 Activity Apoptosis Assay Kit (Geno Technology Inc., St. Louis, MO, USA). 

### 2.10. Measurement the Accumulation of ROS

ROS accumulation was determined using a cellular ROS assay kit (Jiancheng, Nanjing, China), the non-fluorescent 2,7-Dichlorodihydrofluorescein diacetate (DCFH-DA) could be oxidized to fluorescent 2’,7’-dichlorofluorescein (DCF) by ROS. Images were collected using the laser-scanning confocal microscopy (Zeiss LSM 700 META, Oberkochen, Germany), as previously described [24].

### 2.11. Statistical Analysis

Data are presented as the mean ± SEM. Data were analyzed by one-way ANOVA, followed by Fisher’s least significant difference and the statistics analysis was performed using GraphPad Prism version 5.03 (GraphPad Software, San Diego, CA, USA). *p*-values < 0.05 were considered significantly different.

## 3. Results

### 3.1. Overview of Quantitative Proteomic Analyses

Figure 1 shows the workflow of the iTRAQ-based proteomic experiment. In the proteomic analysis, 143,831 unique spectra were strictly matched to 22,236 unique peptides and further mapped to 3022 unique proteins (Appendix A). 

### 3.2. Differentially Expressed Proteins

To analyse the DEPs under different mycotoxins, relative quantification of proteins was performed to analyze the abundance of proteins identified using a threshold of fold changes ≥1.5- or ≤0.67-fold and *p* < 0.05 in different treatments. Compared to the control group, there were 178 abundant DEPs in the β-zearalenol group (Figure 2A), 80 of which were upregulated, including heat shock proteins (HSPs), tubulin proteins, NADH dehydrogenases, and RAB3 GTPase activating proteins, among others. Ninety-eight DEPs were downregulated (Appendix A). In the quantitative analysis of proteins, 291 DEPs were highly abundant in the HT-2 group (Appendix A). There were 119 upregulated DEPs, including tubulin proteins, Ras-related proteins, and HSPs, among others. A total of 172 DEPs were downregulated in HT-2, including collagen, NID2 proteins, and lactadherin. Compared with the differently expressed proteins of β-zearalenol and HT-2, there were 66 common DEPs (Figure 2B; Appendix A).

### 3.3. Analysis of GO and KEGG Pathways for Common DEPs

GO and KEGG enrichment analysis were performed with 66 DEPs. In GO analysis, we significantly enriched 23 biological processes terms (BP) (Figure 3A), 14 molecular functions terms (MF) (Figure 3B), 19 cellular components terms (CC) (Figure 3C), respectively (*p* < 0.05). Moreover, by matching DEPs to the KEGG and NCBI BLAST databases, we enriched six KEGG pathways (*p* < 0.05) (Figure 3D). In the BP terms, most DEPs were assigned to “translation”, “charperone mediated protein”, and “response to unfolded protein”. Moreover, “extracellular exosome”, “extracellular matrix”, and “nucleolus” were the predominant categories in CC terms. In the MF category, most DEPs were assigned to “binding” and “activity”. Figure 3D shows the KEGG pathway enrichment analysis of 66 common DEPs. In the analysis, a total of six pathways were significantly enriched. 

### 3.4. DEPs Analysis

Among these 66 DEPs, nine proteins (seven upregulated) were HSPs (Table 1). The results obtained from the Western blotting analysis indicated that the HSPA5 (GRP78) and HSPB1 proteins were upregulated, and fatty acid synthase (FASN) and hydroxymethylglutaryl-CoA synthase 1 (HMGCS1) were downregulated in both groups (Figure 4). These results showed similar expression patterns as observed in iTRAQ results.

### 3.5. Effect of β-Zearalenol or HT-2 on GC Viability and Apoptosis

Compared to the control, β-zearalenol or HT-2 attenuated the proliferation of bovine GCs in a dose-dependent manner, with 50% inhibition occurring at 25 μM and 50 nM, respectively (Figure 5A,B). According to the cell viability results, doses of β-zearalenol and HT-2 in the following experiment were 25 μM and 50 nM, respectively.

In view of the induction of apoptosis by β-zearalenol/HT-2 in several cell lines, we then examined whether early apoptosis occurred in β-zearalenol- or HT-2-treated GCs. As shown in Figure 5C, the untreated control GCs showed a spontaneous apoptosis rate of approximately 2%, indicating that GCs showed a low baseline rate of apoptosis. However, following the treatment of β-zearalenol or HT-2, the proportion of annexin V-positive cells increased to approximately 12% and 9%, respectively (*p* < 0.01). Moreover, the activity of caspases-3 was also analyzed to investigate the effects of β-zearalenol or HT-2 treatment on apoptosis of GCs. The results indicated that the caspase-3 activities in GCs were markedly increased compared to that of the control group (*p* < 0.01, Figure 5D).

### 3.6. Effects of β-Zearalenol or HT-2 on Estradiol (E_2_) and Progesterone (P_4_) Production in GCs

To further explore the effects of β-zearalenol or HT-2 on the ovarian function, we conducted in vitro experiments to investigate the effects on the production of E_2_ and P_4_. As shown in Figure 6, the content of E_2_ in β-zearalenol or HT-2 group was significantly lower than that in the control group (*p* < 0.01). Moreover, the result of progesterone level revealed no significant difference in the release of P4 between toxin-treated and control groups (Figure 6B).

### 3.7. Effect of β-Zearalenol or HT-2 on GCs ROS

To study the direct effect of β-zearalenol or HT-2 on oxidative state, we analyzed the ROS level in GCs. We found that treatment with β-zearalenol or HT-2 enhanced the ROS accumulation in GCs (*p* < 0.01, Figure 7A,B).

## 4. Discussion

Mycotoxins are toxic substances produced by fungi that are found in various food and feedstuffs, which have acute and chronic toxic effects on several tissues, such as the female reproductive system. [10,13]. Determining the causes/mechanisms of how mycotoxins induce detrimental effects on the reproduction system is important to avoid these adverse effects. The direct ovarian effects of β-zearalenol and HT-2 on pigs have been well documented [25,26,27]. Currently, little information is available regarding the effects of *Fusarium* mycotoxins on ovarian function in cows. Therefore, the purpose of this study was to evaluate the proteomic characterization of *Fusarium* mycotoxin-treated GCs and identify the causes of detrimental effects on reproduction.

In this study, we systematically investigated the GC proteomes in β-zearalenol and HT-2 treatment groups. From the number of differentially expressed proteins, GCs are more sensitive to HT-2 than β-zearalenol (291 vs. 178). Additionally, there are 66 common DEPs between the two groups, which were involved in many biological progress and KEGG pathways. Exposure to β-zearalenol or HT-2 toxins has adverse effects on the biological progresses of translation, protein folding, apoptotic process, and cell–cell adhesion, among others. β-zearalenol and HT-2 toxins affect the pathways of protein processing, biosynthesis, gluconeogenesis, and ribosomes, which may underlie the reduced reproductive potential of livestock and humans. When GCs are exposed to the mycotoxins, a stress response occurs that leads to high expression of the stress protein family [28]. In the biological processes of the GO analysis for the 66 common DEPs, most DEPs were assigned to the “chaperone-mediated protein” and “response to unfolded protein” groups. Our study provided novel information regarding mycotoxin-treated bovine follicles and identified specific highly DEPs in GCs, such as HSPE1, HSPB1, HSPD1, HSPA5, HSPA9, HSPA14, HSP90AA1, and HSPH1 (Table 1). This was further validated by the addition of the Western blot experiment with GCs. HSPs, a family of stress-inducible proteins, their expression, and activity of HSPs are highly regulated by the extracellular microenvironment. The expression levels significantly affect cell survival, proliferation, and apoptosis [29]. Hao et al. reported that HSP expression was fundamental to the reproductive system of mammals, and different HSP members may be linked to follicular development [30]. Our results showed that mycotoxins could induce a change in HSP expression. 

The potential effects of the zearalenone and T-2 toxins on the reproductive activity in pigs are related to the inhibitory effects on GC proliferation and steroidogenesis [26,31]. The high concentrations of ZEA could induce the apoptosis and impair the proliferation of porcine GCs in a dose-dependent manner [32]. T-2 toxin dose-dependently inhibited the growth of GCs and resulted in apoptosis in rat GCs [33]. Our results showed that the effects of β-zearalenol and HT-2 toxins on bovine GCs are related to the activation effects on apoptosis, and the inhibitory effects on cell activity and E_2_ production. In the current study, the apoptosis rate of GCs induced by the β-zearalenol toxin was 12%, whereas apoptosis rate of GCs induced by the HT-2 toxin was 9% after 24 h incubation time. Our results also showed that β-zearalenol and HT-2 toxins impaired the proliferation of bovine GCs in a dose-dependent manner. The progesterone and estradiol secreted by GCs during follicle growth regulate critical phases of the reproductive cycle, such as oocyte growth and maturation [34]. In the current study, β-zearalenol and HT-2 toxins inhibited the production of E_2_ in GCs, which is consistent with the findings of previous experiments [35]. However, there was no change in the level of P_4_ secretion. Our results are not in agreement with findings of Tiemann et al. who used porcine GC and demonstrated that β-zearalenol was able to inhibit P_4_ production [25]. The result was different from that of P4 production that decreased after 2-day exposure to α-zearalenol in bovine large follicle granulose cells [16]. The apoptosis of GCs could partly explain the reason for the decrease of E_2_ secretion [36]. In addition, the change in HSP expression induced by stress could control hormonal functions [37]. In addition, the effects of the P4 level may depend on the presumed target cells, and as such requires further research. 

The increase in mycotoxins-treated accumulation of intracellular ROS levels cause oxidative stress damage and apoptosis [21,27,38], which eventually result in a decline in fertility [39]. The present study demonstrated that β-zearalenol and HT-2 can induce the accumulation of ROS in bovine ovarian granulosa cells. The results are in line with a previous study showing that β-zearalenol and HT-2 trigger oxidative stress in bovine GCs by increasing ROS production [21]. Moreover, ROS production could mediate endoplasmic reticulum (ER) stress [40,41]. To confirm whether ROS mediate ER stress, the relevant marker molecule of ER stress, GRP78, was evaluated. GRP78 plays a significant role in controlling activation of the transmembrane ER stress sensors and protein quality [42]. A considerable amount of work has illustrated that specific changes in GRP78 are indicative of ER stress [43]. GRP78 has been widely used as a marker for the assessment of ER stress. In current studies, the altered expression of the ER stress marker GRP78 (HSPA5) proved that ER stress was involved in β-zearalenol and HT-2 toxin-treated GCs. Our iTRAQ results also showed that the enriched DEPs (66 common DEPs) were involved in protein processing in the ER pathway.

ER stress and cell apoptosis are closely related, ER stress is also involved in GC steroidogenesis [35,44]. There are three key pathways, the mitochondria, death receptor, and ER-mediated apoptotic pathways, involved in apoptosis in mammalian cells [45]. Several studies have reported that caspases are involved in ER stress-induced apoptosis [41,46]. The present study demonstrated that β-zearalenol and HT-2 can induce apoptosis in bovine GCs and increased cleaved-caspase-3 protein expression was observed upon β-zearalenol and HT-2 toxin exposure. These results showed that ER-mediated apoptotic pathways may be involved in β-zearalenol and HT-2 toxin-treated apoptosis in GCs (Figure 8). 

In our study, we determined the effects of β-zearalenol and HT-2 on GCs. However, ZEA and T-2 metabolites can contaminate food and feedstock coincidently; the combined presence of ZEA and T-2 on GCs will require further elucidation.

## 5. Conclusions

The present study characterized the mycotoxin-treated GC proteome of bovine follicles, providing preliminary information that is useful in the elucidation of the clinical effects. These data emphasize the importance of HSPs and clarify ER stress and caspase-3 signaling cascades involved in mycotoxin-treated toxic responses. Our findings also support the hypothesis that β-zearalenol and HT-2 can directly affect the function of GCs and therefore follicular development and ovulation.

## Figures and Tables

**Figure 1 animals-10-00130-f001:**
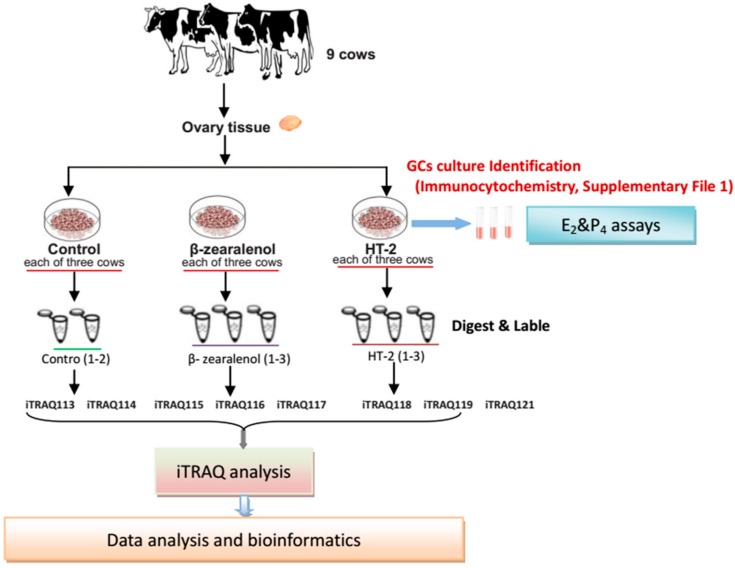
An overview of the experiment. Schematic diagram of the workflow of the isobaric tags for relative and absolute quantification (iTRAQ)-based proteomic experiments. The granulosa cells (GCs) were harvested from nine cows (three cows /group). The cows’ protein samples were pooled for proteomic analyses. For the proteomic assay, the peptides are labelled with different iTRAQ reagents, which contained the control group (113, 114), β-zearalenol group (115, 116, 117), and HT-2 group (118, 119, 121).

**Figure 2 animals-10-00130-f002:**
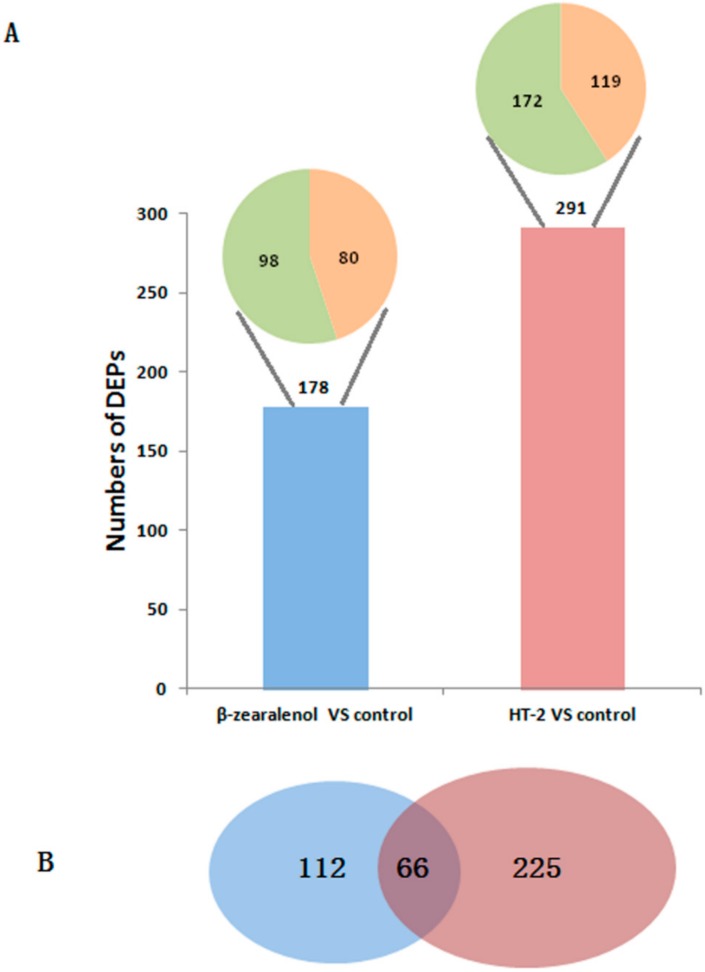
Number of differentially expressed proteins in β-zearalenol and HT-2 groups compared with the control group in GCs. The number above the bar indicates the number of proteins differentially expressed in different treatment groups (**A**). The pie chart represents the differentially expressed proteins (178 and 291), including upregulated differentially expressed proteins (DEPs) (orange) and downregulated DEPs (green). The Venn diagram of different expressed proteins in β-zearalenol (blue) and HT-2 groups (red) demonstrates the overlap between the protein populations in two groups (**B**).

**Figure 3 animals-10-00130-f003:**
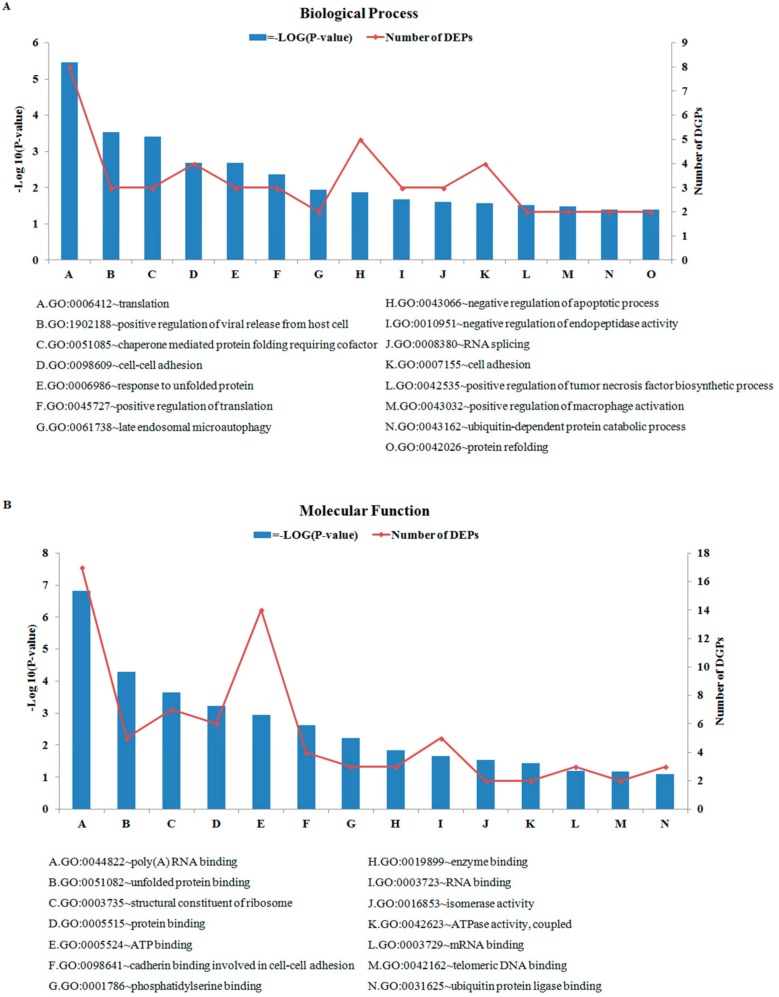
Gene ontology (GO) and Kyoto Encyclopedia of Gene and Genomes (KEGG) analysis of 66 common differentially expressed proteins. Shown above is the classification of these proteins in different categories based on Biological Process (**A**), Cellular Component (**B**), Molecular Function (**C**), and KEGG pathway (**D**). The x-axis shows the functional categories of the increased or decreased proteins, the left y-axis shows the value of –Log (*p*-value), and the right y-axis shows the number of increased/decreased proteins.

**Figure 4 animals-10-00130-f004:**
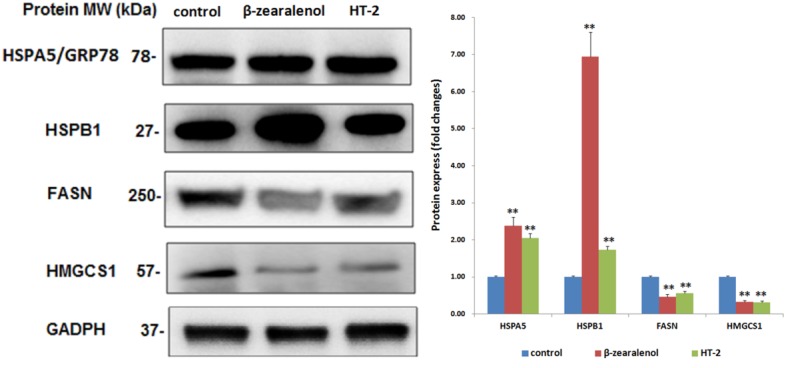
Western blotting analysis of the expression of heat shock proteins (HSPs) (HSPA5 and HSPB1) and other genes (FASN and HMGCS1) in the GCs. GADPH was used as a sample loading control. Values represent the mean ± S.D. ** *p* < 0.01 versus that of the control group.

**Figure 5 animals-10-00130-f005:**
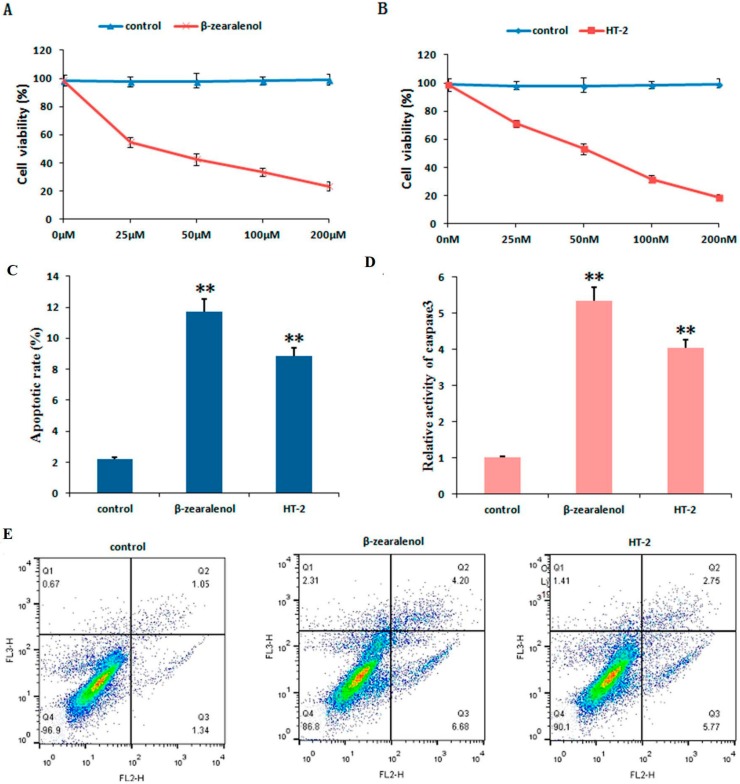
Effects of β-zearalenol or HT-2 on GC viability and apoptosis. The effects of different concentrations of β-zearalenol and HT-2 toxins on the viability of bovine GCs using CCK8 (**A**,**B**). Apoptosis was detected via flow cytometry. After being exposed to β-zearalenol or HT-2 toxin for 24 h, GCs were collected for Annexin V-FITC/PI staining, followed by flow cytometric analysis (**C**,**E**). Cells were treated with β-zearalenol/HT-2 toxin for 24 h, and the cleaved-caspase-3 protein expression level was detected (**D**). Values are expressed the mean ± S.D. of *n* = 3. ** *p* < 0.01 versus that of the control group; CON, control group; β-zol, β-zearalenol treatment; HT-2, HT-2 treatment.

**Figure 6 animals-10-00130-f006:**
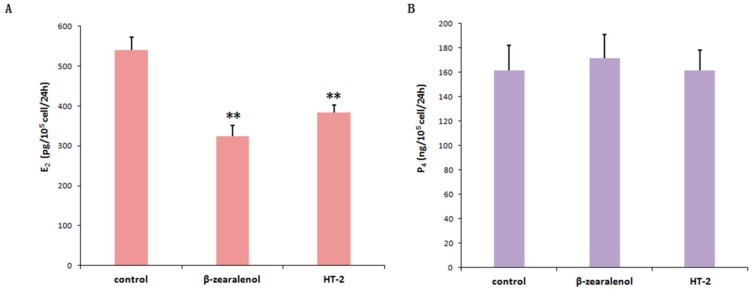
The release of E_2_ and P_4_ in bovine granulosa cells cultured under mycotoxins and a corresponding control over 24 h. (**A**) E_2_ levels in bovine granulosa cell culture medium; (**B**) P_4_ levels in bovine granulosa cell culture medium). Values are expressed the mean ± S.D. of *n* = 3. ** *p* < 0.01 versus that of the control group; CON, control group; β-zol, β-zearalenol treatment; HT-2, HT-2 treatment.

**Figure 7 animals-10-00130-f007:**
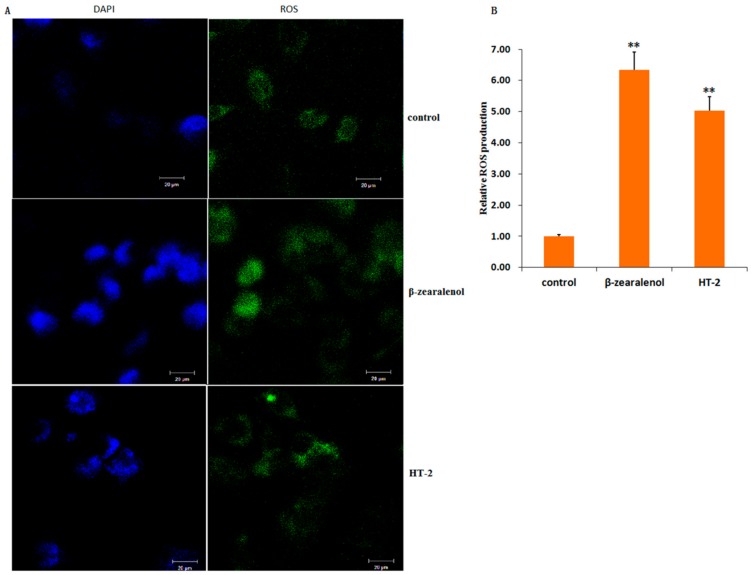
β-zearalenol or HT-2 treated intracellular reactive oxygen species (ROS) accumulation in bovine granulosa cells. (**A**) Fluorescent photomicrographs of GCs stained with 2′,7′-dichlorofluorescin diacetate (H2DCFDA) are shown for control, β-zearalenol/HT-2 over the time course of 24 h. The images shown are representative of the three independent image acquisitions. (**B**) Quantitative analysis of relative fluorescence emission. Values are expressed as the mean ± S.D. of *n* = 3. ** *p* < 0.01 versus the control group; CON, control group; β-zol, β-zearalenol treatment; HT-2, HT-2 treatment.

**Figure 8 animals-10-00130-f008:**
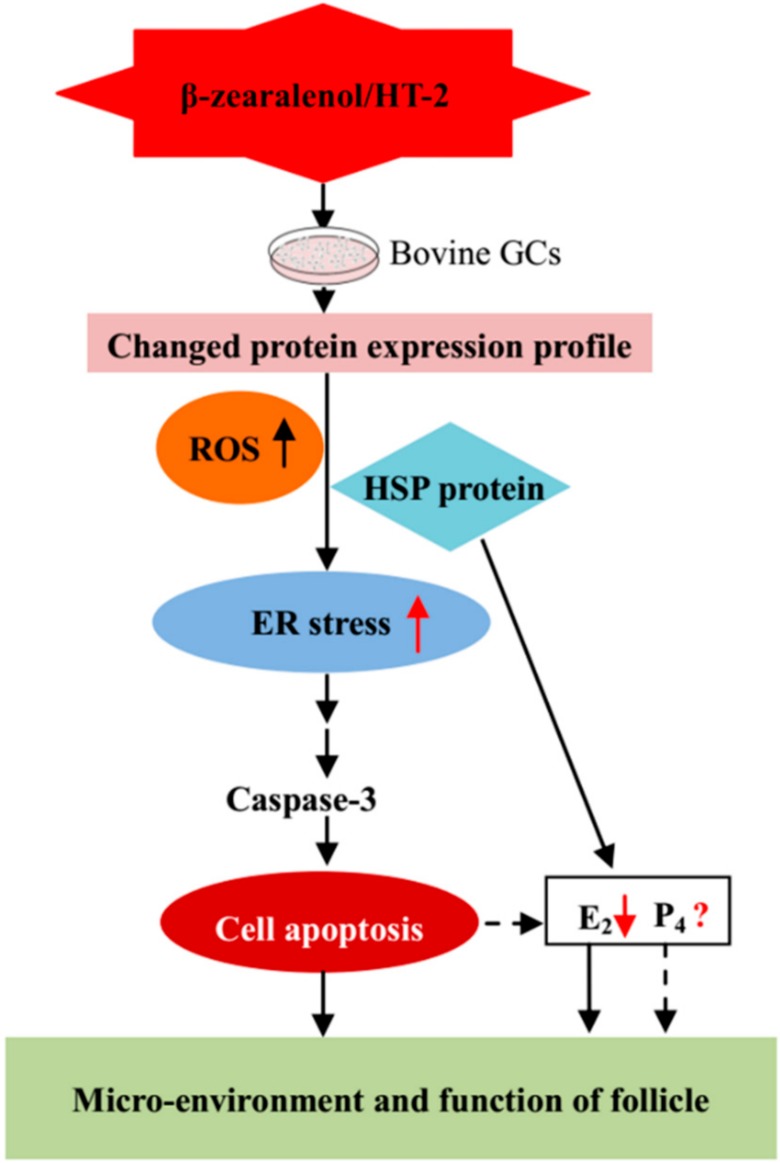
Schematic diagram of the effects involved in β-zearalenol and HT-2 mycotoxin-treated GCs. The proposed effects show that these mycotoxins could cause GC apoptosis, ROS-mediated ER stress, and impair hormone secretion, consequently disrupting the microenvironment and impairing the function of the follicle.

**Table 1 animals-10-00130-t001:** Effects of mycotoxins (β-zearalenol or HT-2) on the expression of selected genes involved in the heat shock protein family (fold changes ≥1.5- or ≤0.67-fold).

HSP Family.	Gene ID	Protein Name	Fold Change		Difference
			β-Zearalenol	HT-2	
HSP10	HSPE1	10 kDa heat shock protein	2.72	2.49	up
HSP27	HSPB1	Heat shock 27 kDa protein	6.79	1.58	up
HSP60	HSPD1	60 kDa heat shock protein	2.17	1.83	up
HSP70	HSPA5/GRP	Endoplasmic reticulum chaperone BiP	1.90	1.82	up
	78 HSPA8	Heat shock cognate 71 kDa protein	3.42	1.95	up
	HSPA9	Stress-70 protein	1.12	1.71	up
	HSPA14	Heat shock 70 kDa protein 14	0.30	0.63	down
HSP90	HSP90AA1	Heat shock protein HSP 90-alpha	1.77	0.48	up/down
HSP105	HSPH1	Heat shock protein 105 kDa	3.69	1.57	up

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
