# Peer review of "Understanding the Toxin Effects of β-Zearalenol and HT-2 on Bovine Granulosa Cells Using iTRAQ-Based Proteomics"

_animals, 2020, doi:10.3390/ani10010130_

Round 1
Reviewer 1 Report
In this manuscript, Li and colleagues use proteomics analysis to unveil the effects of b-zearalenol and HT-2 on granulosa cells physiology. Although the findings are interesting and novel, the authors should do extensive editing to make it suitable for publication. Suggestions to improve the manuscript prior to publication are provided below.
Points to be addressed:
The title requires some modification to make it clearer. An example would be replacing the word “induced” by “treated”. Overall, the manuscript would benefit of extensive grammar review. The effects of mycotoxins on steroidogenesis are not directly elucidated by the proteomic analysis presented in this manuscript. The results of this study point out much more interesting effects on the association between treatments and oxidative stress and apoptosis. It is the opinion of this reviewer that the authors should focus on these and not on steroidogenesis. At the end of the introduction, authors talk about shedding light on steroidogenesis, but no analysis performed during this study directly addresses this point. This should be edited. On the material and methods section, important details are missing. On item 2.1 the diameter of the small follicles and the number of animals used should be stated. On item 2.3, the lysis buffer used should be stated. It is not clear if the authors used cell counting for correction of steroids measurements. If the treatments decrease cell viability, this will result in a decrease in estradiol production due to less cells producing estradiol rather than a direct effect on the steroidogenesis pathway. Therefore, cells from the wells that provide the medium for hormones assay should be collected and counted for correction of steroids per cells numbers. The authors do not mention this in the material and methods section, but show this correction in the figure, which is confusing. This requires clarification. If this was not performed, the authors should consider to eliminate the steroidogenesis measurements of this manuscript and focus on oxidative stress and apoptosis. On section 3.6 of the results, the authors mentioned in vivo experiments, but the material and methods describe in vivo experiments only. This requires revision. On the discussion section (lines 313-315), it is stated that the toxicity of b-zearalenol and HT-2 on follicle function was based on the effect on E2 production of ovarian GCs. However, the correlation of estradiol levels and the results are only observational rather than experimental. So, this statement is highly speculative and should be excluded from the manuscript.Author Response
Thanks for your professional advice. Based on the comments, we made the following changes:
We modified the title-Understanding the toxin effects of β-zearalenol and HT-2 on bovine granulosa cells using iTRAQ-based proteomics. We replaced the word “induced” by “treated”. As you said the research point out much more interesting effects on the association between treatments and oxidative stress and apoptosis. We made some changes to highlight oxidative stress and apoptosis. We add some content about hormones assay in the material and methods section. On section 3.6 of the results: It is our writing error; I have changed in vivo to in vitro. Based on the helpful suggestions and comments, we have revised the manuscript.
More detail information, please see the attachment.

Reviewer 2 Report
The publication submitted for review concerns the proteomic analysis to unravel the effect of β – zearalenol and HT-2 toxin on gene expression and synthesis of steroids in bovine granulosa cells. It is well-known that this substance can harmfully impact living organisms. Our knowledge about the impact of HT-2 and β – zearalenol is still insufficient. The obtained results demonstrate unfavourable effect of toxins on reproductive tract in bovine. Therefore, the results presented in the publication provide an important knowledge about this issue. Unfortunately, I still have found some flaws that require authors' comments. Please refer to the following:
Abstract:
Please delete this sentence: ,, Previous studies have reported 21 that ZEA and T-2 impair mammalian reproduction function” id not appropriate to abstract.
Key words:
I suggest remove reactive oxygen species and add bovine
Introduction:
Line 48 Fusarium use italics.
Moreover, authors should try to explain the practical significance of the obtained results.
Results and materials and methods
How many ovaries did were used in control and in the experimental group?
What phase of the estrus cycle were the animals in? These are very important because of determinate for example sex hormone level in fluid?
Please give catalog number and supplier of follicle-stimulating hormone receptor (FSHR) antibody.
Line 97 remove (Figure 1)
Discussion
Line 274 – 288 contain the same information which are described in results. Moreover, should not be in discussion give information about tables and figures.

Author Response
Thanks for your professional advice. Based on these comments, we made the following changes:
1.Abstract: Please delete this sentence: Previous studies have reported 21 that ZEA and T-2 impair mammalian reproduction function” id not appropriate to abstract.
Response: We delete this sentence.
2.Key words: I suggest remove reactive oxygen species and add bovine
Response: We replace the reactive oxygen species by bovine.
3.Introduction: Line 48 Fusarium use italics. Moreover, authors should try to explain the practical significance of the obtained results.
Response: We revised the format. We rewrite the purpose of study.
4.Results and materials and methods
4.1How many ovaries did were used in control and in the experimental group?
What phase of the estrus cycle were the animals in? These are very important because of determinate for example sex hormone level in fluid?
4.2 Please give catalog number and supplier of follicle-stimulating hormone receptor (FSHR) antibody.
4.3 Line 97 remove (Figure 1)
Response:
4.1 We added some detail information about materials and methods. Please see figure 1 and line 85-100.
4.2 We add the catalog number and supplier.
4.3 We remove the word.
5. Discussion: Line 274 – 288 contain the same information which are described in results. Moreover, should not be in discussion give information about tables and figures.
Response: We rewrite the part.
More detail information, please see the attachment.

Reviewer 3 Report
Dear Editor,
Thank you for sending the manuscript for reviewing. The current study is presenting the effect of two important natural occurring fungal toxins on animal. The in vitro toxicity assessment study was followed by proteomics in order to elucidate the toxic response profile of the studied toxins.
The manuscript needs to be checked by native speaker or by an expert.
Abstract
Line 10 , change “belong to the” to “are two important mycotoxins”
Introduction
Line 39 till 41. Rewrite due to grammatical mistakes.
Line 41 add references … suggested reference >>> A Concise History of Mycotoxin Research. J Agric Food Chem. 2017 Aug 23;65(33):7021-7033. doi: 10.1021/acs.jafc.6b04494. Epub 2016 Dec 27.
Line 42, add the following reference Occurrence of multiple mycotoxins and other fungal metabolites in animal feed and maize samples from Egypt using LC-MS/MS. J Sci Food Agric. 2017 Oct;97(13):4419-4428. doi: 10.1002/jsfa.8293. Epub 2017 Mar 31.
Line 44, add reference >>> suggested reference 1) Occurrence, prevention and limitation of mycotoxins in feeds. And 2) the Occurrence, Toxicity, and Analysis of Major Mycotoxins in Food.
Line 48, all the fungal species should in italic form Fusarium . please double check the whole manuscript.
Line 50, what kind of animal tissues ?? please report.
Material and methods
Line 89, why did you choose these concertations ?
Line 108, change “buffer” to “mobile phase”
Line 111, The separated peptide fragments were identified using a mass spectrometer… HOW ???
The proteomic method is validated ?? published before ?? please give more details about the method. Running time ?? mode of polarity ?? etc
Results
-Figure 2 needs to be provided in high resolution!
-The same for figure 3
-The same for the rest of the figures
Author Response
Thanks for your professional advice. Based on these comments, we made the following changes:
Abstract: Line 10 , change “belong to the” to “are two important mycotoxins”
Response: we have changed it.
Introduction
2.1 Line 39 till 41. Rewrite due to grammatical mistakes.
Response: We rewrite these sentence.
2.2 Line 41 add references
Line 42, add the following reference
Line 44, add reference
Response: We have added these references.
2.3 Line 48, all the fungal species should in italic form Fusarium . please double check the whole manuscript.
Response: We have checked the whole manuscript.
2.4 Line 50, what kind of animal tissues ?? please report.
Response: We have added relevant content.
Material and methods
3.1 Line 89, why did you choose these concertations ?
Response: We have added relevant content
3.2 Line 108, change “buffer” to “mobile phase”
Response: we have changed it.
3.3 Line 111, The separated peptide fragments were identified using a mass spectrometer… HOW ??? The proteomic method is validated ?? published before ?? please give more details about the method. Running time ?? mode of polarity ?? etc
Response: We added some detail information about materials and methods.
Results
-Figure 2 needs to be provided in high resolution!
-The same for figure 3
-The same for the rest of the figures
Response: We modified these figures.
More detail information, please see the attachment.

Round 2
Reviewer 1 Report
The manuscript has been considerably improved.
Author Response
Dear Reviewer,
Thanks for your professional advice.The manuscript have be checked by native speaker(see the attachment).

Reviewer 3 Report
the manuscript is better after considering the comments
line 9, remove "the"
line 129, "the most 20" NOT "the 20 most".
I must highlight that the manuscript needs to be checked for the language but there is not scientific mistakes observed.
Author Response
Dear Reviewer,
Thanks for your advice.The manuscript have be checked by native speaker(see the attachment-certificate of editing).
